# The CHANT’s Conceptual and Psychometric Validity in Switzerland: A Descriptive Three-Round Multicentre e-Delphi Study

**DOI:** 10.3390/nursrep15050141

**Published:** 2025-04-26

**Authors:** Omar Portela Dos Santos, Paulo Jorge Pereira Alves, Henk Verloo

**Affiliations:** 1Department of Nursing Sciences, School of Health Sciences, HES-SO Valais/Wallis, University of Applied Sciences and Arts Western Switzerland, Chemin de l’Agasse 5, CH-1950 Sion, Switzerland; henk.verloo@hevs.ch; 2Faculdade de Ciências da Saúde e Enfermagem, Universidade Católica Portuguesa, Rua de Diogo Botelho 1327, 4169-005 Porto, Portugal; pjalves@ucp.pt; 3Centre for Interdisciplinary Research in Health (CIIS—Wounds Research Lab), Catholic University of Portugal, 4169-005 Porto, Portugal; 4Service of Old Age Psychiatry, Department of Psychiatry, Lausanne University Hospital, Route de Cery 60, CH-1008 Lausanne, Switzerland

**Keywords:** eco-literacy, nursing, nursing sciences, climate change, psychometric validity, content validity index, e-Delphi survey, CHANT

## Abstract

To effectively mitigate the health impacts of climate change, future nurses must be equipped with the requisite knowledge and competencies. Knowing their levels of eco-literacy would help to make them more effective. **Background/Objectives**: This descriptive study will use a three-round, multicentre, modified e-Delphi survey to establish an expert panel’s consensus on the Climate, Health, and Nursing Tool’s (CHANT) item-level and scale-level content validity indices. It will also examine potential associations between the expert panel members’ sociodemographic and professional characteristics and their content validity index assessments of the CHANT. **Methods**: The study will be conducted in the French-speaking regions of Switzerland, running its three-round e-Delphi survey between January and April 2025. After each round, the CHANT’s overall scale-level and individual item-level content validity indices will be computed. Comparisons between different types of healthcare professionals’ profiles will also be conducted. **Results**: The three-round modified e-Delphi survey should allow the expert panel to reach a consensus on the CHANT’s overall content validity index. The tool should then be considered suitable for pilot testing. The first round brought together 16 experts from different regions, namely French-speaking Switzerland, France, and Belgium. **Conclusions**: To ensure that the nursing discipline is well positioned to meet future challenges, the development of eco-literacy must be integrated into nursing education. Ensuring the CHANT’s conceptual and psychometric validity will be essential in strengthening nursing competencies in and knowledge about planetary health and in implementing future educational interventions.

## 1. Introduction

In fewer than 200 years, the dominant models of production and capitalism have led to the widely recognised disruption of Earth’s climate and a collapse in biodiversity [1]. Climate change is a highly complex problem involving many multifaceted and interacting systems [2]. Everybody is a stakeholder. Dutilleul et al. [1] expanded on this by describing and linking the concepts of planetary health and common health. Common health combines all the ingredients and stakeholders needed to create a society in harmony with nature. It aims to be a transformative tool for social and ecological justice through the responsible management and appropriate, equitable and efficient use of natural resources. The concept promotes living within Earth’s capacity for regeneration so that humans can move from a society based on plundering resources, competition and parasitism to one based on commensalism, mutualism and symbiosis. Achieving common health, however, requires a robust society—one with the ability to adapt and remain stable despite fluctuating fortunes [1]. Achieving this requires the societal development of eco-literacy, a term encompassing people’s behaviours, attitudes, practices and knowledge. These four factors are necessary to preserve, protect and take appropriate measures to maintain, restore or enhance ecosystem health and the external conditions influencing human well-being [3].

However, a 2021 World Health Organization survey on adapting to climate change’s impacts on health found that only 52% of countries had an operational national health and climate change plan [4]. This is despite climate change profoundly affecting global health and its recognition as the 21st century’s greatest health threat. Its impacts are wide-ranging, including prolonged heatwaves, extreme weather events, the spread of vector-borne diseases, increased respiratory illnesses due to worsening air quality and the heightened vulnerability of existing disadvantaged populations and regions [5]. Health systems must, therefore, be equipped to effectively address climate-change-related challenges by improving early warning systems for extreme weather events, enhancing healthcare infrastructure to accommodate increased patient loads and ensuring that healthcare professionals are appropriately trained. Indeed, healthcare professionals working at the intersection of patient care and public health are uniquely positioned to mitigate and respond to health issues related to changing climates. Nurses comprise 60% of all healthcare professionals globally and are at the forefront in addressing these health challenges. This makes nursing education a crucial platform in preparing future nurses to engage with and promote Sustainable Development Goal 13 [6].

Nurses are increasingly recognised as key leaders, educators, researchers, advocates and problem-solvers in ongoing efforts to reduce the healthcare sector’s environmental impacts and address the current and future health consequences of climatic, ecological and environmental changes. Universities, thus, have a critical responsibility in preparing nursing graduates and giving them the knowledge and skills needed to tackle the challenges of planetary health effectively [7]. In 2018, the International Council of Nurses emphasised the need to integrate the concepts of planetary health, climate change and sustainability into nursing education to strengthen the profession’s capacity to implement climate-related interventions [6].

From a nursing perspective, addressing the needs of vulnerable populations is imperative and has been a core focus of nursing practice since the time of Florence Nightingale [8]. Nursing interventions should aim to support vulnerable individuals so that they can continue their daily activities, fostering independence and ultimately enhancing their quality of life. Accordingly, care for vulnerable individuals must be personalised, holistic and delivered continuously. The care given to vulnerable populations should not be limited to ensuring physical recovery but should also include psychosocial care and coordination with other community resources.

Individuals who are vulnerable to the effects of climate change experience heightened levels of mental, emotional and physical stress [9]. As defined by Aday [10], vulnerable populations include those at risk of experiencing poor physical, psychological or social health outcomes. They face a heightened risk of developing health problems due to factors such as their marginalised sociocultural status, limited access to economic resources or inherent personal characteristics like age or sex. Vulnerability, therefore, refers to external factors (e.g., environmental) and a lack of capacity to anticipate, cope with and adapt to climate change [11]. The systematic review by Benevolenza and DeRigne [9] underscored the importance of addressing the needs of vulnerable populations, i.e., women, children, older adults and people with chronic illnesses, disabilities or a low income or living in urban neighbourhoods or remote areas [12,13], who often lack adequate coping resources during natural disasters. These groups require tailored, specialised emergency preparedness plans, training programmes and disaster relief measures to ensure their safety and well-being. Climate change vulnerability refers to the likelihood of being adversely affected under conditions of risk. It can be examined through three main components: sensitivity, exposure and adaptive capacity. The degree of vulnerability differs across populations based on the interactions between these factors. According to Cheng et al. [13], vulnerability indicators can be divided into three categories: (1) socioeconomic status and cultural background (age, income, education, living alone or in a group, ethnicity, population density, employment, language proficiency, place of birth, home ownership, sex, family structure, job specification, vehicle ownership, having a home or being homeless, living in an urban area and social class); (2) health status (healthcare or caregiving services, morbidity and mortality); and (3) the environment (climate data, living environment and housing conditions) [13].

Nurses play a uniquely valuable role in responding to the health challenges faced by the patients who are the most disproportionately affected by climate change. They must also contribute to environmental justice, which is described as the right to equal protection from environmental and health hazards and an equitable opportunity to create, implement and enforce environmental laws, regulations and policies, regardless of race, ethnicity, national origin or income [14]. The barriers to and facilitators of the implementation of environmentally sustainable practices in healthcare can be categorised at the individual, institutional, geographical, infrastructural and political levels. At the individual level, barriers include nurses’ limited knowledge or skills regarding the topic, time constraints, competing priorities, diverse interpretations, moral offset, concerns about increased workloads, feelings of powerlessness and denial, resistance to change and fear of the legal consequences related to reusing certain instruments. Conversely, key enablers include nurses’ engagement, motivation, perceived benefits and access to continuing professional education [15].

Climate change is a complex, multidimensional issue involving numerous interdependent systems and stakeholders. Moreover, certain uncertainties persist, particularly regarding its real impacts, the regions that will be most affected and the timescales over which these effects will manifest [2]. Ambiguity and conflict avoidance suggest that the general public struggles to comprehend information involving uncertainty and is less inclined to take or endorse precautionary measures when risks are characterised as uncertain. Studies conducted in the United States have demonstrated that the perceived scientific consensus on climate change is linked to support for climate mitigation policies. However, this relationship is mediated by individuals’ beliefs and concerns about climate change. The stronger the belief that climate change is occurring and driven by human activity, the greater the willingness to act [16,17]. Moreover, people with more perceived knowledge tend to report less concern for climate change, while those with more assessed knowledge tend to report more concern [16]. Assessing nurses’ levels of eco-literacy and then implementing educational interventions is necessary to enable them to take a positive stance on climate change and act accordingly.

Eco-literacy refers to the behaviours, attitudes, practices and knowledge that maintain, restore or improve the ecosystem and all the external conditions that affect human health [18]. The study by Kotcher et al. [19] found that 41% of the 4654 healthcare professionals surveyed felt that they lacked adequate knowledge to effectively educate patients and the public, despite most of them recognising climate change as a moderate to significant threat to public and patient health. Additionally, nearly 30% expressed scepticism about their role in addressing climate change, believing that their engagement would have little impact. Different studies have indicated that nurses possess only moderate levels of eco-literacy, whether in professional or personal settings. Moreover, their perceptions of climate change and its health effects were generally weak or insufficient [20,21,22]. Although nurses recognised the health impacts of climate change—such as air pollution (66.7%), inadequate waste management (45.7%), heatwaves (37.2%), climate change itself (27.9%) and flooding (25.7%)—particularly on vulnerable populations such as older adults and children [20], 55.2% remained uncertain about the specific positive actions that they could take [23]. These findings align with those of Schenk et al. [22], who surveyed 487 nurses using the Climate, Health and Nursing Tool (CHANT). This revealed moderate levels of awareness (M = 2.97; SD not reported; min–max: 0–4) but high levels of concern (M = 3.43; SD not reported) about the health consequences of climate change. Several barriers to nurses’ engagement with these issues were also identified, including limited knowledge and a sense of being overwhelmed by the topic’s complexity. Consequently, the number of reported actions addressing climate change in workplace settings remained low. Badawy [24] explored nurses’ lived experiences, motivations, challenges and perspectives, concluding that they played a critical dual role in promoting eco-consciousness, both as educators and advocates. Engagement in educational initiatives and advocacy for eco-consciousness must extend beyond direct clinical duties and help to foster a culture of environmental stewardship in healthcare. To enable nurses to promote planetary health and adopt an appropriate stance on this issue, they must develop the appropriate knowledge and skills and be exposed to evidence-based research and collaborations that contribute to improving the environmental health of all populations. This can be achieved by increasing awareness among nurse educators, integrating climate-change-related content into nursing curricula, promoting interdisciplinary and multidisciplinary approaches, strengthening clinical practice skills and fostering advocacy and leadership capabilities [25]. However, the first step must be to identify the levels of knowledge and awareness about climate change and climate-associated diseases—nurses’ eco-literacy.

A systematic review published in 2024 [26] evaluated the most reliable, robust and valid instruments for the measurement of nurses’ knowledge and awareness of climate change and climate-associated diseases. Among the eight instruments included, the CHANT, developed by Schenk et al. in 2019, presented the best psychometric properties [27]. The CHANT measures nurses’ awareness, motivation, concerns and self-reported behaviours at work and at home. One of its principal strengths is that, despite being developed for nurses, it can be used to measure the cognitive behavioural levels of other healthcare professionals [22,27].

The CHANT is designed to be completed in approximately 10–12 min and comprises six subscales: awareness, experience, concern, optimism, motivation and behaviour. It consists of 22 items assessed using five-point Likert scales. Awareness is measured by evaluating knowledge about climate change’s effects on health using a score ranging from 1 (“Not at all familiar”) to 5 (“Extremely familiar”). One item also explores the sources consulted for information on climate change. Experience is assessed through four items scored from 1 (“Not at all familiar”) to 5 (“Extremely familiar”). Worry about climate change is evaluated using five items that measure levels of worry, scored from 1 (“Not at all”) to 5 (“Extremely”). Optimism regarding mitigation and adaptation is assessed by two items scored from 1 (“Not at all”) to 5 (“Extremely”). Motivation to support mitigation and adaptation efforts is measured by three items scored from 1 (“Never”) to 5 (“Always”). Finally, behaviour is analysed through two sets of items. Five items assess ecological actions taken at home and four evaluate those implemented at work. Both sets of items are scored from 1 (“Never”) to 5 (“Always”) [22,27].

The translation of the CHANT followed Wild’s ten-step, structured best practices [28], ensuring both linguistic accuracy and cultural appropriateness. The preparation phase clarified key concepts, obtained approvals and recruited local translators (D.H.) and consultants (H.V. and P.J.P.A.). Approval for translation was granted via email by Prof. Schenk on 21 October 2024. Translation from English to French was performed independently by two people (O.P.D.S. and P.J.P.A.). Any discrepancies were addressed in collaboration with HV, a native French speaker who had not been involved in the translation. A French-to-English back-translation was then conducted by a local translator (D.H.). O.P.D.S., H.V. and P.J.P.A. conducted a revision process to identify and correct any discrepancies. Two rounds of harmonisation were performed to ensure consistency between the French and English versions. A cognitive debriefing involving five registered nurses was conducted to assess the tool’s clarity and cultural relevance in French. No adjustments were needed. Finally, proofreading addressed any remaining grammatical errors, and a final report documented the entire process to support interpretation and future translations [28]. The next step involves its comprehensive adaptation to Switzerland’s cultural context to ensure conceptual equivalence between the original and adapted versions as part of the translation and cultural adaptation process. The two research questions are the following:What is the expert panel’s consensus on the CHANT’s item-level content validity index (I-CVI) and scale-level content validity index (S-CVI)?Are there any significant associations between the expert panel’s sociodemographic and professional characteristics and their assessments of the CHANT’s I-CVI and S-CVI scores?

The careful cross-cultural adaptation of measurement instruments is essential in ensuring their reliability, validity and relevance in diverse cultural contexts [29,30]. This process goes beyond mere translation and involves comprehensive cultural adaptation that preserves the conceptual equivalency between the original and adapted versions [29]. Ensuring this approach’s success requires the consideration of multiple types of equivalence. Conceptual equivalence assesses whether the domains measured are relevant to the target culture, and item equivalence examines the alignment between the questions and the intended concepts [30]. Semantic equivalence ensures an accurate and meaningful translation, operational equivalence verifies the appropriateness of measurement methods, and metric equivalence evaluates the comparability of psychometric properties across versions [30]. However, cultural biases can pose significant challenges [29]. Methodological bias may arise from differences in response styles, question formats or social norms, whereas content bias occurs when certain items are inappropriate or unclear to the target culture [29,30]. Additionally, construction bias emerges when the concept being measured does not hold the same meaning across cultures, affecting its comparability. A rigorous methodological approach is essential in identifying these potential biases early and correcting them, thus ensuring that the adapted instrument remains both culturally appropriate and psychometrically robust [29,30].

## 2. Methods

The expert panel’s consensus decisions will guide our investigation of the CHANT’s content validity as expressed by the understandability of the tool’s items and their associated response options, the overall scale’s validity information and adjustments for potential chance agreements. To best achieve our research objectives, the study was designed to use the well-established e-Delphi technique [31] and aligned with Junger et al.’s [32] 2017 guidance on Conducting and Reporting Delphi Studies (CREDES). The e-Delphi method was chosen for its flexibility, its iterative nature and its structured group facilitation approach that systematically gathers expert insights while ensuring asynchronous participation and participant anonymity. Among the different Delphi types [33], we chose a modified e-Delphi design that could be administered via an online internet survey platform, SurveyMonkey. Whereas traditional and modified Delphi methods gather expert panel feedback through a series of structured rounds, a real-time e-Delphi survey uses specialised software to conduct a ‘roundless’ process, allowing for the continuous and dynamic exchange of information among panel members within a specified timeframe [31]. The e-Delphi survey will be carried out in several stages, including the recruitment of expert panel members, the administration of the survey itself, data analysis and the reporting of findings. Before initiating the e-Delphi study, it was critiqued and piloted by the three research team members to address the criteria involving transferability.

### 2.1. Sampling

Selecting suitable members for the expert panel is critical to a Delphi study. As Sablatzky noted [34], although the panel does not need to be representative of a broader population, its participants must be engaged, well-informed and knowledgeable about the subject. Their expertise, perspectives and potential biases can significantly influence the study’s outcomes. To ensure credibility, confirmability and consensus in the present study, we will recruit expert panel members who fulfil at least one of the following criteria: (1) they have good knowledge about planetary health, climate change and sustainability; (2) they live in French-speaking Switzerland, France or Belgium; or (3) they work in the healthcare sector. The criteria are broad because of the benefits of having a diverse sample regarding demographic characteristics and professional experience. Indeed, diverse participants bring different skills and points of view, thus ensuring that opinions come from multiple independent sources [35].

Participants will be recruited via email and LinkedIn using a convenience sampling method. While there is no fixed sample size for an e-Delphi panel, it is generally accepted that including at least 10–18 members enhances the reliability of the group’s judgments [36]. Non-probability snowball sampling will be used so that our experts can transfer the CHANT’s internal evaluation questionnaire to other experts in their professional circle.

### 2.2. Data Collection

e-Delphi cycles will be conducted via email using the online platform SurveyMonkey. Experts will receive an email outlining the study’s objectives and inviting them to rate the questionnaire’s items on a Likert scale from 1 (“Strongly disagree”) to 4 (“Strongly agree”). They will be encouraged to provide written feedback to enhance each item’s relevance to the domain at hand, to evaluate the clarity of the wording and to identify any ambiguities, redundancies or unnecessary statements. They will also be invited to suggest improved wording or propose the inclusion of additional knowledge or skill statements. All feedback will be carefully considered to refine the tool in subsequent cycles. The questionnaire’s first page will include details about the study, its estimated completion time, a data confidentiality notice and the process for publishing results.

The study will consist of three iterative evaluation cycles, each informed by the synthesis of the responses from the previous round to enable progressive refinements. Each cycle will last four weeks, with reminder emails sent out two weeks before each deadline to maximise participation. The survey’s methodological characteristics will enable honest responses, promoting dependability and credibility.

### 2.3. Data Analysis

A consensus will usually be determined by the percentage of agreement with a given response, followed by the proportion of participants who rate items at the highest levels of the Likert scale used (e.g., scores of 4 and 5 on a five-point Likert scale) [35]. e-Delphi studies generally set an a priori level of consensus, which can range from 51% to 100%, with the median acceptable threshold for consensus being 75% agreement among the participants [37].

The data collected will be input into an Excel^®^ spreadsheet (www.microsoft.com, Redmond, WA, USA) (accessed on 1 January 2025), and two experienced data managers (O.P.D.S. and H.V.) will inspect it for errors, missing values and consistency. This will then be imported into the Statistical Package for the Social Sciences (SPSS) software, version 29.0 (IBM-SPSS Inc., Chicago, IL, USA), for analysis. Descriptive statistics will be used to characterise participants’ sociodemographic and professional profiles. Measures of the central tendency and dispersion (means and standard deviations) will be calculated for quantitative variables, while frequency distributions and percentages will be used for qualitative variables. To assess the stability of the responses between rounds two and three, we will compute the kappa coefficient and its 95% confidence interval based on the respective percentages of group agreement for each category (i.e., the proportion of participants who rated the category as “relevant” in each round). Content validity will be assessed quantitatively using the Content Validity Index (CVI), calculating both the item level CVI (I-CVI) and the overall scale level CVI (S-CVI) [38,39]. The I-CVI represents the proportion of experts who score an item at 3 or 4 on a five-point Likert scale, with a threshold of 0.78 set as the criterion for acceptability [39]. Items with an I-CVI below 0.78 will be revised or eliminated from the questionnaire, based on the experts’ recommendations, to ensure its overall validity and clarity. The S-CVI will be determined using the average method (S-CVI/Ave), which calculates the mean I-CVI across all items. According to Polit and Beck [39], the acceptability threshold is ≥0.90. The second method is the universal agreement method (S-CVI/UA), which measures the proportion of items scored as relevant by every expert (I-CVI ≥ 0.78). The acceptability threshold is ≥0.80 [38,39]. In addition to the quantitative assessment, qualitative expert feedback will be analysed to refine or reword items identified as unclear or insufficiently relevant. The research team will rewrite or adapt items that do not reach a consensus in rounds one and two by having two authors (O.P.D.S. and H.V.) perform a content analysis of the experts’ comments using open coding. Finally, the third and final e-Delphi round will consist of a cognitive debriefing to discuss and finalise the agreement about the CHANT’s understandability and to validate its overall IVC.

## 3. Ethical Considerations

Voluntary participation in the multi-step modified e-Delphi survey process will be considered as consent to participate, although the experts will be informed that they can withdraw at any moment. Data anonymisation, personal anonymity and the controlled feedback process provided during all three rounds will guarantee the process’s confidentiality and transparency.

## 4. Results

The first round of the modified e-Delphi survey successfully engaged a panel of 16 experts. The panel demonstrated diverse geographical representation, including professionals from the French-speaking region of Switzerland, France and Belgium. This geographic and professional diversity will contribute to the richness and contextual relevance of the data collected. Moreover, the inclusion of experts from various health systems and educational environments will enhance the transferability and credibility of the consensus process within similar sociolinguistic and healthcare settings.

## 5. Discussion

This study’s three-round, multicentre, modified e-Delphi survey process is designed to facilitate the formation of a consensus by an expert panel regarding the CHANT’s S-CVI. Iterative rounds of assessment, feedback and refinement will enable our experts to evaluate the tool’s relevance, clarity and comprehensiveness. Once a satisfactory level of agreement has been reached, the tool will be considered valid and appropriate for pilot testing in real-world settings to further assess its applicability, reliability and effectiveness in measuring nurses’ eco-literacy and awareness of climate-change-related effects on health.

## 6. Conclusions

It is crucial that today’s nurses be equipped with the knowledge and skills needed to address the growing global health issues brought about by climate change. Guaranteed future planetary health and worldwide common health currently elude humanity, even though their fate rests solely in its hands. A robust, adaptable society is the key to responding to the effects of climate change. Indeed, this robustness enables populations to strike a balance between people’s need for health and well-being in peaceful, equitable societies and their need to live in and with a natural environment that is treated respectfully [1].

To align Switzerland with the stance taken by the International Secretariat of French-Speaking Nurses or the American Nurses Association’s Principles of Environmental Health for Nursing Practice, it has become crucial to measure nurses’ knowledge and awareness of climate change and climate-change-associated diseases. To do so, it will be necessary to translate and comprehensively adapt the Climate, Health and Nursing Tool (CHANT) to Switzerland’s cultural context. We believe that only through this process can nursing expertise be recognised as a driving force for change in Switzerland, enabling the profession to integrate interventions to improve the social and environmental determinants of health into its action plans. This approach should foster collaboration with local, regional and national decision-makers, allowing nurses to actively participate in assessing the quality of their practice and patients’ living environments. It should also enable them to contribute to research on best practices that promote safe, healthy environments and to receive support in advocating for and implementing environmental health principles in nursing practice. By emphasising the importance of integrating environmental health into both initial and continuing nursing education—by establishing nurses’ eco-literacy—this study will aid the development of targeted educational interventions to enhance nurses’ competencies in this area. This approach contributes directly to Sustainable Development Goal 13 while promoting care practices that are more aligned with the current climate challenges. Finally, providing nurses with well-structured training modules on sustainability and climate action is essential to address existing challenges effectively.

## Data Availability

Data are contained within the article.

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
