# Peer review of "The CHANT’s Conceptual and Psychometric Validity in Switzerland: A Descriptive Three-Round Multicentre e-Delphi Study"

_nursrep, 2025, doi:10.3390/nursrep15050141_

Round 1
Reviewer 1 Report
Comments and Suggestions for Authors
Strengths of the study (my opinion)
- The study follows the well-established e-Delphi method, ensuring a structured, iterative process to achieve expert consensus. Aligning with CREDES guidelines adds credibility.
- The asynchronous participation feature allows experts from diverse locations and time zones to contribute without scheduling constraints. Anonymity minimizes groupthink and social desirability bias.
- The inclusion of experts with backgrounds in planetary health, climate change, sustainability, and healthcare enhances the validity and applicability of findings.
- Multiple Evaluation Rounds: The three-round process ensures progressive refinement, improving the clarity and relevance of questionnaire items.
- Quantitative and Qualitative Analysis: Combining Likert scale responses, statistical validation (CVI, kappa coefficient), and qualitative feedback analysis strengthens the study’s validity and reliability.
- Use of Established Statistical Tools: The use of SPSS for analysis ensures robust statistical processing and enhances the study’s credibility.
Weaknesses of the study (my opinion)
- Sampling Limitations:
Convenience and Snowball Sampling: These non-probability sampling methods can introduce selection bias, potentially limiting the generalizability of the findings.
Panel Size Uncertainty: I am little bit worry how many participans is going to take part into research. While there is no fixed sample size, relying on a minimum of 10–18 experts might not capture the full spectrum of opinions in a highly specialized field.
- Potential for Response Bias:
Given that participation is voluntary and the recruitment is via email/LinkedIn, experts who are more engaged or have strong opinions might be overrepresented.
Asynchronous communication, while beneficial for flexibility, may limit real-time debate that could clarify ambiguities in the questionnaire.
- Data Quality Concerns:
Relying on self-reported ratings via online surveys may result in variability in the interpretation of the Likert scale, despite the provided guidelines.
The iterative nature of the survey could lead to participant fatigue over multiple rounds, potentially affecting the quality of later responses.
- Technical and Logistical Considerations:
The reliance on a single online platform (Survey Monkey) may be limiting if technical issues arise, potentially disrupting the flow of data collection.
Coordinating multiple rounds over several weeks requires careful management to ensure timely participation and consistency in data collection.
Author Response
"Please see the attachment."

Reviewer 2 Report
Comments and Suggestions for Authors
As a protocol for a future study, the proposal submitted is formally adequate. There is an appropriate theoretical presentation in which the relevance of the study context and its importance for the specific group towards which the instrument to be investigated would be oriented is given. The methodology is appropriate for the proposed objectives, as it allows the evaluation of the validity of the content of the CHANT.
However, since it is not yet possible to observe the results of the study, it is not possible to assess its overall merit, the interest it could generate and its true importance for the target audience. I believe that the submission of the article with results of the procedure it describes will allow a correct weighting of the contribution of the article.
Author Response
"Please see the attachment."

Reviewer 3 Report
Comments and Suggestions for Authors
Quantitative results should be presented using tables or graphics, and should be discussed in relation to the findings from other reviewed literature.

Author Response
"Please see the attachment."

Round 2
Reviewer 2 Report
Comments and Suggestions for Authors
As a protocol, the article provides an adequate description of the procedures that are committed to be carried out in the next stages of evaluation of the psychometric properties of CHANT in Switzerland. In this sense, it presents an adequate presentation of the most important literature on the subject and the methodological procedures are clearly described. However, it is difficult to make a complete judgment regarding its relevance without having its results and conclusions in view. With this information, there is no doubt that it will be able to contribute adequately with respect to an instrument that will account for a contingent theme such as the understanding of changes in subjective dispositions oriented towards the phenomenon of climate change, specifically in nurses.
Author Response
We sincerely thank you for the relevant feedback, which helped improve the quality and relevance of our development and usability pilot study protocol.
Reviewer 3 Report
Comments and Suggestions for Authors
I do not agree with your response because you should describe the characteristics of the experts and participants in the results section. Additionally, you used SPSS for descriptive statistical analysis and employed a rating scale questionnaire, so you should present the results for each item. Moreover, surveying participants and experts involves interfering with their normal lives, which makes it impossible to avoid seeking ethical approval for research involving human subjects.

Author Response
"Please see the attachment."
